# LOB-Bench: Benchmarking Generative AI for Finance - with an Application to Limit Order Book Markets

## Abstract

We present **LOB-Bench**, a benchmark designed to evaluate the quality and realism of generative message-by-order data for limit order books (LOB). We enable a rigorous and comprehensive model comparison by providing both a theoretical framework and an open-source Python package. Addressing the lack of consensus on evaluation paradigms in the literature, where qualitative comparison of stylized facts is prevalent, our work offers a crucial building block for advancing generative AI for financial data. LOB-Bench provides a standardized method to numerically assess the quality of various model classes that generate limit order book data in the widely used LOBSTER format. It provides a range of quantitative characteristics and includes a simple parametric benchmark model as a baseline. Our framework measures distributional differences in conditional and unconditional statistics between generated and real LOB data, supporting a flexible multivariate statistical evaluation across different model classes. The benchmark features commonly used LOB statistics such as spread, order book volumes, order imbalance, and message inter-arrival times, along with adversarial scores derived from a neural network trained to differentiate between real and generated data. Additionally, LOB-Bench evaluates "market impact metrics" by computing cross-correlations and price response functions for specific events in the data. We present empirical benchmark results for a generative autoregressive state-space model, for a (C)GAN, and parametric LOB model. We find that the autoregressive GenAI approach beats traditional model classes. All our code and example generated data is available at: https://github.com/anon-ml-review/lob_bench_review.

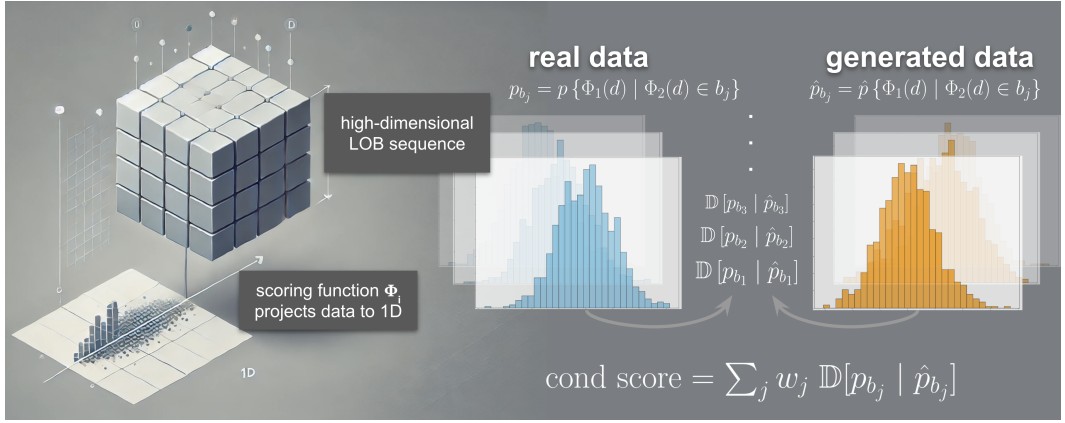

Figure 1: Schematic of LOB-Bench methodology for conditional distributional evaluation

## 1 INTRODUCTION

Generative AI (GenAI) is currently revolutionizing different fields, ranging from natural language processing to image generation and real world applications. Perhaps surprisingly, the backbone of all of these methods is simply self-supervised pre-training on large datasets using a next-token prediction loss on auto-regressive sequence models. (Nie et al. (2024); Dubey & et. al. (2024); Liu et al. (2024))

Recently, Nagy et al. (2023) applied this paradigm to *limit order books*, i.e. the mechanism through which stock markets keep track of buy and sell orders to determine any-time prices. Specifically, in contrast to prior work, which models only high level features , this approach learns a *token-level* distribution over messages in the LOBSTER dataset (Huang & Polak (2011)).

In principle, an *accurate*, *low level* generative model of the financial system would be extremely valuable from a societal and commercial point of view. For example, it could unlock better mechanism design, stability analysis, or learned-order execution (Frey et al. (2023)) through answering "what if" questions, i.e. providing counterfactuals.

A key question then is how to determine the realism and trustworthiness of GenAI, and of other generative financial models. On the one hand, for high-level approaches and "old school" agent-based modeling Byrd et al. (2020); Chiarella & Iori (2002); Paulin (2019); Llacay & Peffer (2018) the evaluation is usually based on a qualitative analysis of whether the model reproduces known high-level patterns (e.g. "stylized facts") from the literature, such as "impact" or the famous "square-root law" (Tóth et al. (2016); Brokmann et al. (2015); Almgren et al. (2005b)). However, this is not a quantitative or general evaluation.

On the other hand, for GenAI the standard evaluation for pre-training is simply *cross-entropy*, i.e. how closely the model is able to predict the next token on held-out data. Unfortunately, this does not capture how the model performs under autoregressive sampling, when *generating* sequences of data one token at a time, where error accumulation can cause distribution shift. In many applications of GenAI this is not a problem, since the pre-trained models are merely used as *starting points* for task specific fine-tuning (e.g. RLHF), rather than in their "bare" form. In contrast, we want to evaluate the pre-trained models in the *sampling* regime to unlock the mentioned use-cases.

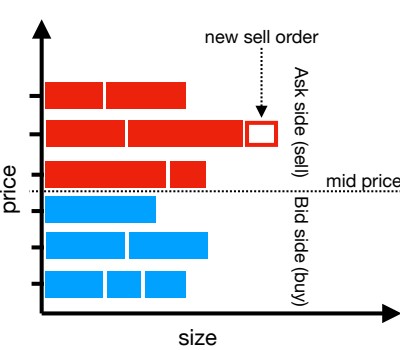

Figure 2: Schematic of the LOB.

To address this, we propose a general framework for evaluating the similarity between the distribution induced by financial GenAI models and the ground truth data. At a high level, our *unconditional* evaluation consists of three steps. We first introduce a set of *aggregator functions*, $\Phi$, which map from high dimensional time-series LOB data into a set of 1d subspaces. Secondly, we compute histograms for the ground-truth and generated data in these subspaces and, finally, use a distance metric, e.g. L1, to compare the 1d histograms. Some of the aggregator functions chosen are closely inspired by metrics used in literature, such as spread, orderbook imbalance etc. Vyetrenko et al. (2021); Paulin (2019); Chiarella & Iori (2002); Cont (2001). They also directly relate to *generative adversarial networks*, where the discriminator network is equivalent to a *worst-case* aggregator function for a given generator.

For *conditional* distributional evaluation, we first apply an aggregator function to group the data into "buckets" based on the conditioning variable. We then score each of the resulting conditional distributions using the process described earlier. This approach enables, for example, assessing whether the distribution of bid-ask spreads, conditioned on the time of day, aligns with the corresponding conditional distribution in real data. As another example, we can evaluate whether a discriminator-based score reveals that generated sequences are easier to distinguish from real data at specific times of day. To derive a single metric, we compute the average loss across the conditioning buckets, weighted by the probability of each bucket. Furthermore, we can also use this to evaluate model-drift by aggregating on the *sampling step* and comparing to the unconditional data, which is a good proxy for model-derailment in open-loop sampling. See Figure 1 for a process schematic.

We test our evaluation framework on three different generative models: two state-of-the art GenAI models (Coletta et al. (2022); Nagy et al. (2023)) and a widely-used classic model as a baseline Cont et al. (2010). All models are tested on data of Alphabet Inc (GOOG) and Intel Corporation (INTC) stock. We don't present detailed results for the Coletta model trained on INTC, because this was developed only for small-tick stocks and fails on INTC data Coletta et al. (2022). We find evidence of "model derailment", since the distance scores increase for longer unrolls. However, for some scoring functions, this might be partially attributed to the fact that our generated sequences are "seeded" from true data as models are initialized by providing an initial book state before generation starts. We also find that our framework is mostly able to reproduce the standard *price-impact curves* that are well-known in the economics and finance literature Eisler et al. (2012). See section 6 for details.

Finally, there are features which are not directly measurable on the ground truth dataset, since they require counterfactuals but are well established in literature. In contrast, generative models allow to directly evaluate counterfactuals, so in the future we plan to measure to what extent it matches the perhaps most famous one of these, the "square root law" (SRL) of market impact Tóth et al. (2016).

Our contributions are summarized as follows:

- **A novel LOB benchmark for distributional evaluation:** We introduce the first LOB benchmark focused on full distributional quantification of model performance. This addresses limitations of prior work, which relied on qualitative comparisons of stylized facts, making rigorous model comparisons infeasible and hindering research progress.

- **Interpretable scoring functions for targeted improvements:** using intuitive scoring functions enables targeted model development and refinement.

- **Difficult challenge of discriminator scores:** discriminator-based scoring sets a high bar for future generative models, even when most other statistics are closely aligned.

- **Identification of a common failure mode:** divergence metrics, computed as distributional errors as a function of unroll step, highlight a prevalent failure mode that can guide research.

- **Ease of use and accessibility:** The benchmark is open-source, straightforward to apply and only requires data in the LOBSTER format.

- **Extensibility:** LOB-Bench can be easily extended to additional scoring functions.

- **Transferability to other domains:** The underlying theoretical framework is adaptable to other high-dimensional generative time-series tasks beyond LOB data.

We hope our benchmark will provide a much-needed starting point for evaluating GenAI models in finance and allow more machine learning scientists to develop new sequence models for this important and challenging domain. Our code is available at the following link: https://github.com/anon-ml-review/lob_bench_review.

## 2 BACKGROUND

### 2.1 LIMIT ORDER BOOK (LOB)

Later sections of this paper rely on the reader's understanding of the mechanisms of electronic markets, so we briefly review them here. Public exchanges such as NASDAQ and NYSE facilitate the buying and selling of assets by accepting and satisfying buy and sell orders from multiple market participants. The exchange maintains an order book data structure for each asset traded. The limit order book (LOB) represents a snapshot of the supply and demand for the asset at a given time. It is an electronic record of all the outstanding buy and sell limit orders organized by price levels. A matching engine, such as first-in-first-out (FIFO), also called *price-time* priority, is used to pair incoming buy and sell order interest as mentioned in (Bouchaud et al. (2018)). Order types are further distinguished between limit orders and market orders. A limit order specifies a price that should not be exceeded in the case of a buy order (bid), or should not be gone below in the case of a sell order (ask). A limit order queues a resting order in the LOB at the corresponding side of the book. Placing a limit order at a price level is sometimes referred to as placing a quote. A market order indicates that the trader is willing to accept the best price available immediately, see Figure 2 for an illustration.

In real-time trading, injecting orders into the market induces other market participant activity that typically drives prices away from the agent. This activity is known as market impact (Almgren & Chriss (1999); Almgren et al. (2005a)). Presence of market impact in real time implies that a realistic trading strategy simulation should include deviation from historical data. Therefore, realistic market impact emulation is an important consideration in limit order book modeling.

## 2.2 LOB MODELS

LOB simulation is an important technique for evaluating trading strategies and testing "what if" market scenarios. The extent to which results from such simulations can be trusted depends on how accurately they emulate real world environments. In the past literature, it is common to use historical market data for trading strategy training and backtesting and to make an assumption of negligible market impact, given the size of agent orders is small and a sufficient amount of time is allowed between consecutive trades (Spooner et al. (2018)). However, the "no market impact" assumption is not valid for larger order sizes. Agent-based methods naturally allow to study such phenomena, which emerge as a consequence of multiple participant interactions, which are difficult to model otherwise. However, they are notoriously challenging to calibrate (Vyetrenko et al. (2021)). To circumvent calibration, conditional generative adversarial networks were used to learn simulators from historical LOB data, that are both realistic and responsive (Coletta et al. (2023)). Most recently, an end-to-end autoregressive generative model that produces tokenized LOB messages in the spirit of generative AI was shown to achieve a high degree of realism (Nagy et al. (2023)).

## 2.3 AUTOREGRESSIVE LOB MODELS

In machine learning, autoregressive modeling is a key component of language models like GPT. By learning the probability distribution of the next token given the previous tokens, autoregressive language models can generate coherent text (Radford et al. (2019)). Cross-entropy is a loss function commonly used to train classification models in deep learning. It measures the dissimilarity between the predicted class probabilities and the true class labels (Goodfellow et al. (2016)). Cross-entropy loss is the negative log likelihood of the true class labels under the predicted distribution. Minimizing the cross-entropy is equivalent to maximizing the likelihood of the data (Murphy (2012)). For binary classification, the cross-entropy loss is:

$$L = -\frac{1}{N} \sum_{i=1}^{N} [y_i \log(\hat{y}_i) + (1 - y_i) \log(1 - \hat{y}_i)]$$

where $y_i$ is the true label (0 or 1) and $\hat{y}_i$ is the predicted probability for the positive class. Cross-entropy loss heavily penalizes confident misclassifications and incentivizes the model to output calibrated probabilities that match the empirical distribution of the classes. Although it is different from the KL divergence, cross-entropy can be expressed as the sum of the entropy of the true distribution and the KL divergence between the true and predicted distributions (Cover & Thomas (1999)).

## 3 RELATED LITERATURE

Limit order books (LOBs) play a crucial role in modern financial markets. Numerous studies focus on using LOB data for mid-price prediction and market impact analysis. With the FI-2010 dataset, Ntakaris et al. (2018) released the first publicly available high-frequency LOB dataset for benchmarking mid-price prediction models. This dataset contains tick-by-tick order data for five stocks on the Nasdaq Nordic market for ten consecutive days, standardized for machine learning tasks. Although useful and effective for preliminary tests and comparisons of LOB algorithms, FI-2010 does not allow a comprehensive evaluation of robustness and generalisation ability (Zhang et al. (2019)). A similar benchmark for average price and volume prediction in Chinese stock markets is provided by Huang et al. (2021). Similarly to other currently available benchmarks, this work falls short of evaluating GenAI models with a fully distributional lens. Cao et al. (2022) propose a benchmark dataset, which plays a complementary role to LOB-Bench. With DSLOB, they provide a synthetic LOB dataset, generated by a multi-agent simulation with shocks, which generates labeled in- and

out-of-distributions samples. In contrast, LOB-Bench does not require training on a specific dataset, and instead focuses on general-purpose model evaluation and comparison.

To evaluate the performance of generative models in the LOB environment, several studies have proposed relevant metrics. Coletta et al. (2023) investigated the interpretability, challenges, and robustness of conditional generative models. They grouped LOB states based on certain attributes and statistics and then performed conditional generation on these groups. Vyetrenko et al. (2021) proposed several statistics to assess the realism of LOB simulators, such as order arrival rate, order distance distribution, and price volatility.

In summary, although some studies have addressed the evaluation of generative models for LOBs, a unified benchmarking framework is still lacking. Existing research often uses *qualitative* methods to compare statistical regularities of generated data with real data, lacking quantitative evaluation metrics. Therefore, establishing a comprehensive benchmarking framework for evaluating LOB generative models is essential for advancing the field.

## 4 EVALUATION FRAMEWORK

As the success of LLMs has shown, generative models can already achieve impressive performance by autoregressive training, or "next-token prediction" alone. However, not all model classes are auto-regressive or allow the explicit computation of conditional "next-token probabilities," prohibiting cross-entropy based evaluation or calculating model perplexity Chen et al. (1998). However, there is still a need to evaluate such model classes, where we can merely sample data. Another reason why single-token cross-entropy loss is insufficient is the so-called "autoregressive trap" (Zhang et al. (2024)). Even small errors in a next-token prediction task can accumulate over long sequences, thereby moving the data away from the training distribution. Out-of-distribution forecasts then become increasingly worse until the generating data distribution could completely derail or collapse. This emphasizes the need to evaluate statistics of entire sequences, rather than focusing solely on cross-entropy. This also implies that a benchmark framework should measure how fast such errors accumulate by evaluating distributions conditional on the forecasting horizon.

Evaluating generative models in any domain is fundamentally a matter of comparing distributions. Our benchmark performs exactly this task. It reduces a high-dimensional distribution of sequences of order book states $\mathbf{b} \in \mathcal{B}$ and message events $\mathbf{m} \in \mathcal{M}$ to scalars by using scoring functions $\Phi_i : (\mathcal{M} \times \mathcal{B}) \mapsto \mathbb{R}$, $i \in \mathbb{N}$. One-dimensional score distributions can then be compared between real and model-generated data using various norms or divergences $\mathbb{D}$. By estimating the difference between the unconditional real data distribution $p\{\Phi(d)\}$ and the data distribution under the model $\hat{p}\{\Phi(d)\}$,

$$\mathbb{D}\left[p\left\{\Phi(d)\right\} \mid\mid \hat{p}\left\{\Phi(d)\right\}\right], \tag{1}$$

different generative models can be ranked on their ability to match features of the data.

To evaluate the magnitude of the "autoregressive trap" the benchmark evaluates error divergence of distributions, conditional on the interval of the forecasting step $t \in \mathbb{N}$, for interval limits $a, b \in \mathbb{N}$: $\mathbb{D}\left[p\left\{\Phi(d_{t\in[a,b)})\right\} \mid\mid \hat{p}\left\{\Phi(d_{t\in[a,b)})\right\}\right]$. This allows quantifying distribution shift during inference.

Our framework uses both the L1 norm and the Wasserstein-1 distance as loss metrics. To estimate the L1 norm we first bin the data. As a robust binning algorithm, we use the Freedman-Diaconis rule (Freedman & Diaconis (1981)), which computes the bin width as $2\frac{IQR}{\sqrt[3]{n}}$, where n is the combined sample size of the real and generated data. The $[0, 1]$-scaled L1 norm, also called total variation distance, can then be estimated as:

$$\frac{1}{2}\left\|p - \hat{p}\right\|_1 = \sum_{b \in bins} \frac{1}{2} \mid p(b_{count}/b_{width}) - \hat{p}(b_{count}/b_{width}) \mid . \tag{2}$$

While the L1 measure has the benefit of being bounded in the interval $[0, 1]$, the Wasserstein-1 distance, or earth mover's distance, as proposed in Rubner et al. (2000), has the advantage of being sensitive to the distance between the scores. To make losses between different scoring functions comparable, we mean-variance normalize the data before calculating the Wasserstein-1 distance. The L1 distance is conceptually simple and is proportional to the area of mismatched bins between histograms of both distributions and is therefore an intuitive measure of distributional similarity.

For equal sample sizes we can compute the Wasserstein-1 distance as follows. Let $\Phi(d_{real})_{(i)}$ be the i-th order statistic of a score computed from a real data sample drawn from $p$ and $\Phi(d_{gen})_{(i)}$ the i-th order statistic using generated data drawn from $\hat{p}$. Then we have:

$$W_1(p, \hat{p}) = \sum_{i=1}^{n} \left\| \Phi(d_{real})_{(i)} - \Phi(d_{gen})_{(i)} \right\|_1. \tag{3}$$

To evaluate a generative model's ability to adapt to different contexts, we also estimate differences between conditional score distributions

$$\mathbb{D}\left[ p\left\{ \Phi_1(d) \mid \Phi_2(d) \right\} \| \, \hat{p}\left\{ \Phi_1(d) \mid \Phi_2(d) \right\} \right]. \tag{4}$$

In this case, $\Phi_2(d)$ is binned into 10 data deciles $b_j$ of the pooled real and generated data. Distance estimates of these 10 conditional distributions are then weighted according to the mean of the estimated density of both distributions. Letting $X = \Phi_1(d)$ and $Y = \Phi_2(d)$, we have

$$\sum_{b_j} \mathbb{D}\left[ p(X \mid Y \in b_j) \| \, \hat{p}(X \mid Y \in b_j) \right] \frac{p(Y \in b_j) + \hat{p}(Y \in b_j)}{2}. \tag{5}$$

This approach enables addressing a specific type of distribution shift: the variation of scores, $\Phi_1$, across the distribution of another score, $\Phi_2$. For instance, if the conditioning function $\Phi_2$ represents the mean time of messages within a data sequence, this framework allows us to analyze how distribution shifts affect any score of interest, $\Phi_1$, and to assess the generative model's ability to replicate this dynamic behavior accurately.

## 4.1 IMPACT RESPONSE FUNCTIONS

A primary difficulty with data sets of limit order book data is that counterfactual scenarios are impossible to evaluate. This is because historical data is, by definition, static and will not respond with market impact to any additional injected orders. Generative models of synthetic LOB data are, therefore, a unique opportunity to generate a response to counterfactual scenarios as new data may be generated given different conditional inputs.

When building generative models it is therefore crucial that they be evaluated on their ability to provide a realistic response to different events. As an underlying methodology, the seminal work by Eisler et al. (2012) is used as a basis to compare the impact of different event types. This methodology focuses only on the impact of events, which change the price or quantity of the best bid and ask orders (sometimes also referred to as the touch orders), which is concurrently one of its limitations.

All events which affect the best prices are classified into one of six order types $\pi \in \Pi$: market orders (MO), limit orders (LO) and cancellations (CA), which are further subdivided into those which affect the mid-price, indicated with subscript 1, and those who do not, with subscript 0: $\Pi = \{MO_0, MO_1, LO_0, LO_1, CA_0, CA_1\}$.

Following the convention used in *LOBSTER* data, we define the direction ($dir$) as 1 for events on the bid side and $-1$ on the ask side. The events are given an $\epsilon$ value based on the expected direction of impact on the mid-price they will provoke. Notably, there are no market order events in the *LOBSTER* datasets, but rather execution events that match orders on the opposite side of the book. For such the epsilon values have a switched sign, as with cancel orders:

$$\epsilon = \begin{cases} dir & \text{if event type is MO or LO;} \\ -dir & \text{if event type is CA.} \end{cases} \tag{6}$$

The key function of interest for comparing real and generated data is the response function (equation 7). This is calculated empirically using the time average ($\langle \quad . \quad \rangle_T$) of the change in the sign-adjusted mid-price $p_t = \frac{a_t + b_t}{2}$ following a given event, for different lag-times $l$. The event lag times are chosen to be distributed uniformly on a logarithmic scale between 1 and 200 ticks. The prices are normalized by tick size to enable a comparison between various stocks.

$$R_\pi(l) = \langle (p_{t+l} - p_t)\epsilon_t \mid \pi_t = \pi \rangle_T \tag{7}$$

Eisler et al. (2012) identify averaged response functions for 14 random stocks over a period of 53 trading days. Whilst such analysis gives a good baseline to which we can compare our results, for model evaluation we instead directly compare the functions between model-generated and real sequences (following the same preceding "seed" sequence) for individual stocks. Once the response functions are calculated, we create a measure of comparison to obtain a score of dissimilarity:

$$\Delta R_\pi = \frac{1}{L} \sum_{l=1}^{L} |R_\pi^{real}(l) - R_\pi^{gen}(l)|, \tag{8}$$

which is aggregated across all event types by taking the mean $\Delta R = \frac{1}{|\Pi|} \sum_{\pi \in \Pi} \Delta R_\pi$.

## 4.2 ADVERSARIAL MEASUREMENT

The concept of adversarial measurement is to develop a pre-trained discriminator capable of effectively distinguishing between true and generated trajectories. This discriminator is a binary classifier, generating a probability estimate of a trajectory being real. The input to the discriminator is a sequence of orderbook states. The discriminator is trained on two batches of data, each of dimension $(S \times T \times D)$. In this representation, $S$ denotes the number of sequence samples within the batch, $T$ the length of the orderbook sequences, and $D$ is the dimension of the orderbook state representation. Given the sparsity of changes between most orderbook states, we devised an encoding scheme to optimize the discriminator's performance.

The discriminator aims to find the "worst-case" function $\Phi^*$ that maximally separates the real and generated distributions by choosing $\Phi^*$ such that it maximizes the divergence between them, i.e., $\Phi^* = \arg\max_\Phi D[p(\Phi(d)), \hat{p}(\Phi(d))]$. This worst-case $\Phi^*$, which can be interpreted as a dimensionality reduction operation from a high-dimensional data distribution of a sequence of order book states $\mathbf{b} \in \mathcal{B}$ and message events $\mathbf{m} \in \mathcal{M}$ to a scalar $s$, $\Phi^* : (\mathcal{M} \times \mathcal{B}) \mapsto \mathbb{R}$, can also be thought of as an adversarial scoring function. For a given generator, the discriminator seeks to learn the function that results in the highest possible loss for the generator. In other words, it tries to identify the most glaring flaws and differences between the real and generated samples.

An orderbook state comprises the price and quantity from the top $n$ price levels on both the bid and ask sides. For instance, selecting the top 10 price levels would result in an orderbook state with 40 dimensions, evenly split between the bid and ask sides. However, changes in the orderbook state are typically triggered by events that affect a single price-quantity pair. To achieve a more concise, yet informative, representation of the discriminator network, we chose to represent the orderbook based on these changes. Thus the book states $\mathbf{b} \in \mathcal{B}$ and message events $\mathbf{m} \in \mathcal{M}$ map to three-dimensional vectors through $i \in \mathbb{N}$ functions $\Psi_i : (\mathcal{M} \times \mathcal{B}) \mapsto \mathbb{R}^3$. These changes encompass each change in the mid-price, the relative price level where the change occurs, and the corresponding change in quantity. Our discriminator utilizes a 1D convolutional neural network (Conv1D) (LeCun et al. (1995); Kiranyaz et al. (2019)) as a feature extractor, followed by an attention mechanism (Vaswani et al. (2017)) to capture long-term dependencies across the time steps. Empirical results show that this model, trained and tested on GOOG data from 2023, achieves a Receiver Operating Characteristic (ROC) score of 0.83, indicating that the generated data can be discriminated reasonably accurately. However, the baseline model's performance for GOOG and INTC was poor, with a discriminator ROC score of around 1, indicating significant room for future model improvement.

## 5 LOB-BENCH PACKAGE

Based on the evaluation framework outlined in section 4, we developed a Python benchmark package, allowing for a convenient and comprehensive evaluation of generated LOB data. The benchmark is highly customizable as scoring functions $\Phi$ can easily be added, removed or modified, and provides a standardized model comparison using the provided default scoring functions. The benchmark reports aggregate model scores by computing the mean, median, and inter-quartile mean (IQM[1]) across all conditional and unconditional scoring functions, along with bootstrapped confidence intervals.

The benchmark performs both unconditional and conditional evaluation of generated data, by computing distributions of statistics of interest conditionally on the value of another statistic. To evaluate the

---

[1]mean of all values between the 25. and 75. percentile

magnitude of the effect of error divergence or "snowballing errors," distributions are also evaluated conditional on the prediction horizon. Distributional accuracy is measured by computing the L1-norm and Wasserstein-1 distance between the real and generated distributions. Specific supported examples of more complex conditional distributions are market response functions, describing the distribution of events conditional on other events having occurred at a certain prior lag. As these distributions usually have high variance, and to be consistent with the extant literature, we instead measure mean absolute differences in their means for a range of lags to evaluate market impact curves.

We include multiple conditional scoring functions from the finance literature, for example, ask volume conditional on the spread, the spread conditional on the hour of the day, and the spread conditional on the volatility of 10ms returns. This two-dimensional slicing of score distributions evaluates the adaptability of generative models to different market scenarios or contexts.

| Statistic | Description |
|---|---|
| Bid-Ask Spread | Difference between the highest price a buyer is willing to pay (the bid) and the lowest price a seller is willing to accept (the ask). |
| Order Book Imbalance | The LOB imbalance for the best prices is computed as (bid size − ask size)/(bid size + ask size). |
| Message Inter-Arrival Time | The time between successive order book events, evaluated on a log-scale due to a long right tail. |
| Time-to-Cancel | For limit orders, which are canceled before execution, this is the time between submission and first (partial) cancellation. Due to a long right tail, this is measured on a log-axis. |
| Bid/Ask Volume | The volume of all orders on the bid, respectively ask, side of the LOB. We also evaluate the volume only at the best price levels. |
| Bid/Ask Limit/Cancellation Depths | Absolute distance of new limit orders or cancellations from the mid-price. |
| Bid/Ask Limit/Cancellation Levels | The price levels at which events occur $\in \mathbb{N}$. |

The benchmark also evaluates model response functions (equation 7) in aggregate. Individual L1 distances $\Delta R_\pi$ are calculated for each lag time and averaged to produce aggregate impact scores.

## 6 RESULTS

As a test case for our benchmark, we have adapted the autoregressive state-space model using S5 layers (Gu et al. (2021)) from Nagy et al. (2023) (*LOBS5*). Particularly, we have scaled up the model size by 10x in the number of parameters and more than doubled the training period to the entire year of 2022. Furthermore, for this larger model, we successfully removed the explicit error correction mechanism, which originally rejected semantically incorrectly generated messages. To illustrate the use of our benchmark we trained two separate models on Alphabet (GOOG) and Intel (INTC) stock, in line with Nagy et al. (2023).

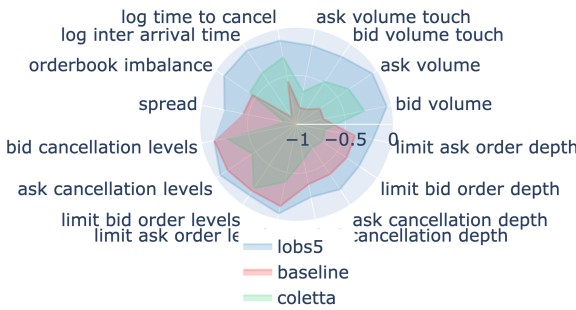

Figure 3: Model comparison spider plot: the *LOBS5* model beats the *baseline* and *Coletta* model on almost all scores. Note: the radial axis is inverted by plotting the negative loss (larger is better).

We also evaluated data generated by the models from Cont et al. (2010) (*baseline*) and Coletta et al. (2022) (*Coletta*). The baseline model, which employs parametric arrival processes, was adapted to generalize across both small and large tick limit order book (LOB) dynamics by utilizing estimated empirical arrival rates directly, rather than fitting a power law. Additionally, we inferred data features present in *LOBSTER*, such as individual message IDs, which were not generated by Cont et al. (2010). This inference is particularly important for capturing order cancellations, as we uniformly sample target limit orders from the available orders at the specified price level. For the *Coletta* model, we implemented a *LOBSTER* data interface to facilitate the conversion of data formats. All

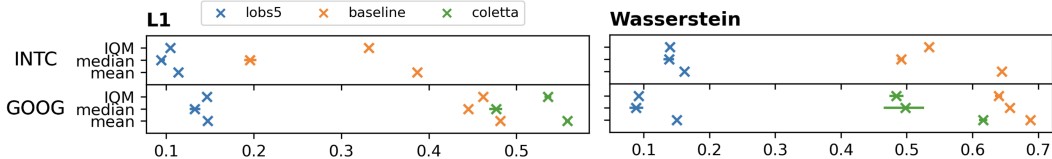

Figure 4: Model score summaries (lower is better). The *LOBS5* model achieves the lowest overall scores. *coletta* beats the *baseline* on the Wasserstein metric, but not for L1. Error bars are bootstrapped 99% CIs.

results presented here were computed on a sub-sample of the test data from January 2023, except the *Coletta* model, which was trained on three days from January 2019 and tested on three subsequent days, following the procedure in Coletta et al. (2022), necessitates by the high computational cost of training and running inference for *Coletta*. Comparing all models, we note that the *LOBS5* model provides state-of-the-art performance on the benchmark task.

Figure 3 demonstrates a key benchmark feature to compare multiple models across multiple score dimensions, allowing a critical examination of individual strengths and weaknesses. To provide summary scores per model, Figure 4 reports the mean, median, and inter-quartile mean for the L1 and Wasserstein-1 metrics for all available models [2]. Error bars demarcate the 99% bootstrapped confidence intervals. Metrics for individual scoring functions are shown in Figure 8 in the appendix.

The benchmark also measures error divergence by comparing distributions of scoring functions, conditional on the inference time step. These demonstrate the rate at which distributions diverge from real data. Results show increasing errors across all models with the fastest divergence exhibited by the baseline model. Scoring functions with a strong dependence on features of the generated book states, which only gradually change, such as book volume, are expected to produce increasingly worse results, as the initial real data seed decays. However, the rate of decay can still be compared between models. See Figure 12 in the appendix for L1 divergence curves.

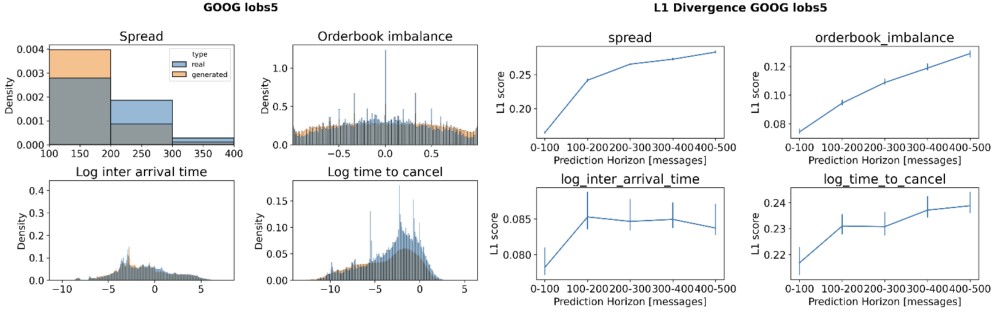

Figure 5: *LOBS5* results - (left): histogram matching of unconditional score distributions for real and generated data. (right): Error accumulation - the further out the prediction horizon, the worse is the model performance - an important model characteristic to measure.

The response functions for Alphabet (GOOG) are shown in Figure 7. The *LOBS5* model generally reproduces curves similar to real data, but does so better for small-tick stock GOOG. In contrast, the *baseline* model Cont et al. (2010) is unable to faithfully reproduce impact curves. Confidence intervals for some lag values are particularly large for the generated Intel (INTC) data. Likely, this is also incited by the sparsity of the book, complemented by the relatively infrequent occurrence of market orders, which affect the best prices. The L1 distance between the real and generated curves (equation 8) and their averages are **0.099** and **0.105** for GOOG and INTC respectively, with the biggest contributors being the distances for $MO_1$ and the $CA_1$ orders. One reason for the difference in $MO$ orders at short lags is due to the treatment of the JAX-LOB Frey et al. (2023) simulator at

---

[2]The *Coletta* model Coletta et al. (2022) was trained on both GOOG and INTC data, but failed to produce reasonable results for INTC, which is explainable as it was intended for small-tick stocks, which INTC is not.

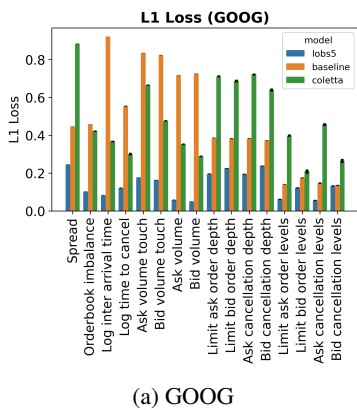
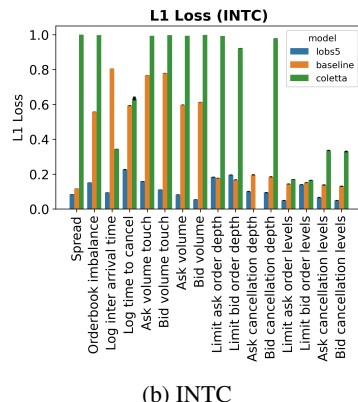

(a) GOOG                                          (b) INTC

Figure 6: L1 distance between real and generated data histograms (99% bootstrapped CIs). The *baseline* performs well on LOB depth and level-related scores, and much worse on time and volume metrics. *LOBS5* dominates L1 loss for GOOG, and dominates the L1 loss for INTC for most scores.

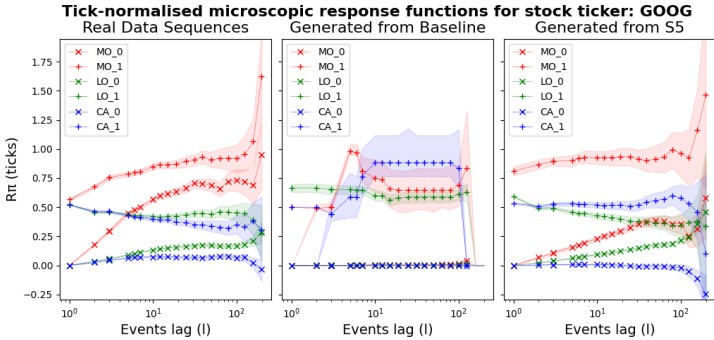

Figure 7: Comparison of impact response functions for different event types between real and generated data-sets, tick-normalized mid-price response. Shaded regions are 99% confidence intervals. There is a comparison between two select models: the LOBS5 and the stochastic baseline. We see that, in contrast to the baseline, the generative model is able to reproduce most of the expected impact response function.

inference time, as used by the *LOBS5* model. This interprets some large messages as execution and an additional limit order, merging a multi-level mid-price change into a single order book update.

## 7 CONCLUSIONS

We introduce LOB-Bench, an evaluation framework for generative AI models for order-book modeling. Crucially, our framework contains analysis tools that make it easy for users across the machine learning and finance domain to benchmark their message-level order-flow models.

We believe that LOB-Bench will greatly facilitate core ML research working on sequence modelling to apply their innovations to this challenging and relevant real-world problem and will also make it easier for finance practitioners to use best-practice tools.

One of the interesting aspects of generative AI models for microstructure data is the ability to model counterfactuals which closely relates to the notion of price impact in financial modelling. In conventional approaches it is highly challenging to factor in the reactions of other market participants to one's actions. Within our benchmark suite for generative LOB models, we provide extensive tests to evaluate that the generated data reproduces the expected response functions at a larger scale, which is highly non-trivial. We hope that this opens the door to many new studies, including training of reinforcement learning algorithms and multi-agent models for trade execution with the ability to model realistic reactions of different market participants.

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

# A APPENDIX

## A.1 BENCHMARK CODE

The benchmark code can be found on GitHub at https://github.com/anon-ml-review/lob_bench_review.

The benchmark suite provides a convenient API functionality to evaluate model data for a range of scoring functions and metrics. A specification of such functions and loss metrics can be defined in a configuration dictionary, which can then be passed to function performing the unconditional and conditional model evaluation. Similarly, the benchmark provides functions to compute the market impact curves, along with a mean L1 score. A default configuration dictionary, specifying the scoring functions reported here, evaluated using L1 and Wasserstein-1 loss, is similarly provided for easy reproducibility.

To run the benchmark, real and generated data sequences must be stored in LOBSTER format [3] as csv files. Files must be separated by real data, generated data, and (real) data used to condition the generation. A more detailed description can be found on GitHub.

## A.2 ADDITIONAL FIGURES

---

[3]https://lobsterdata.com/info/DataStructure.php

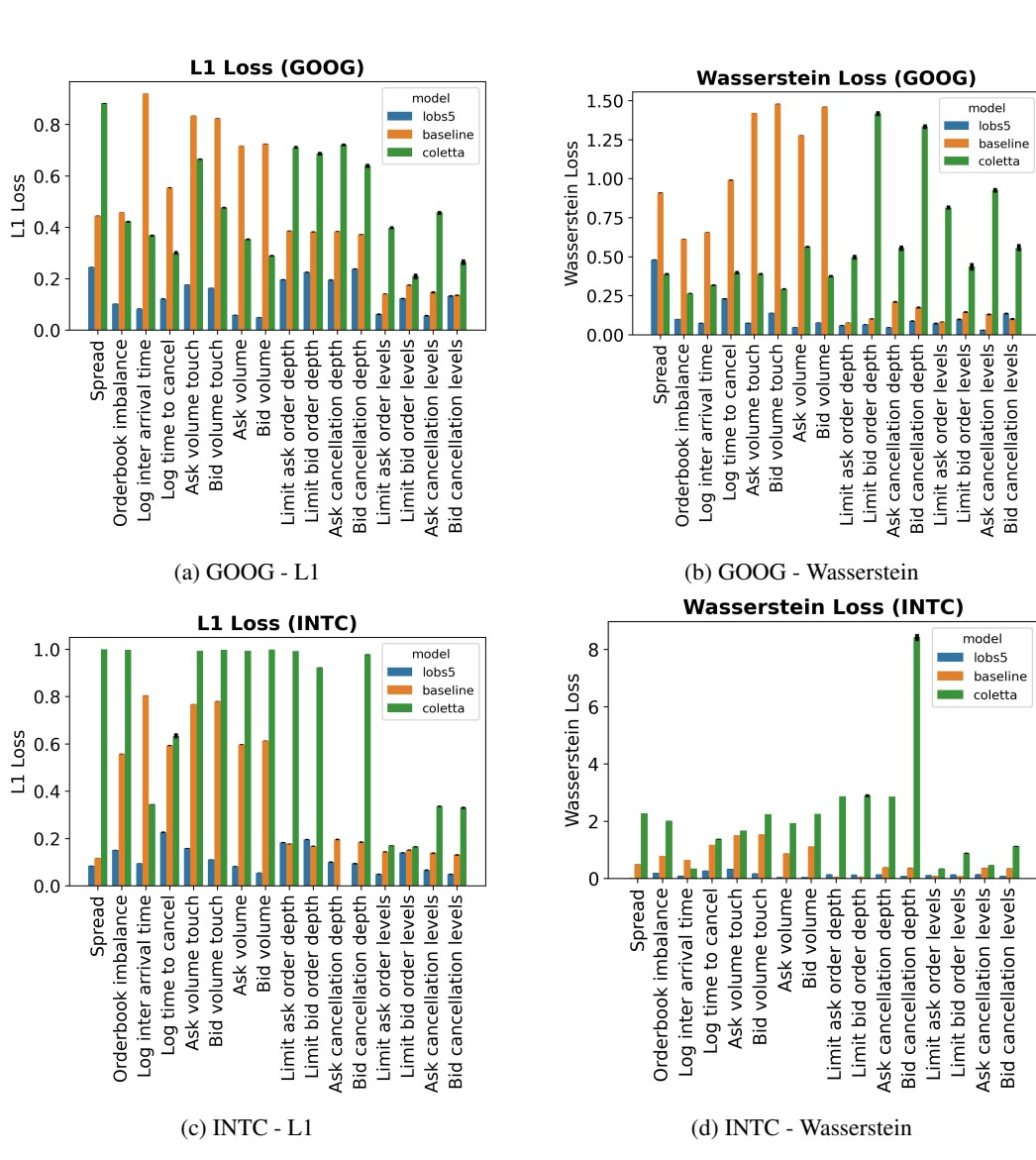

Figure 8: L1 and Wasserstein-1 errors of generated unconditional distributions for easy comparison between Alphabet (GOOG) and Intel (INTC). Error bars show 99% bootstrapped confidence intervals.

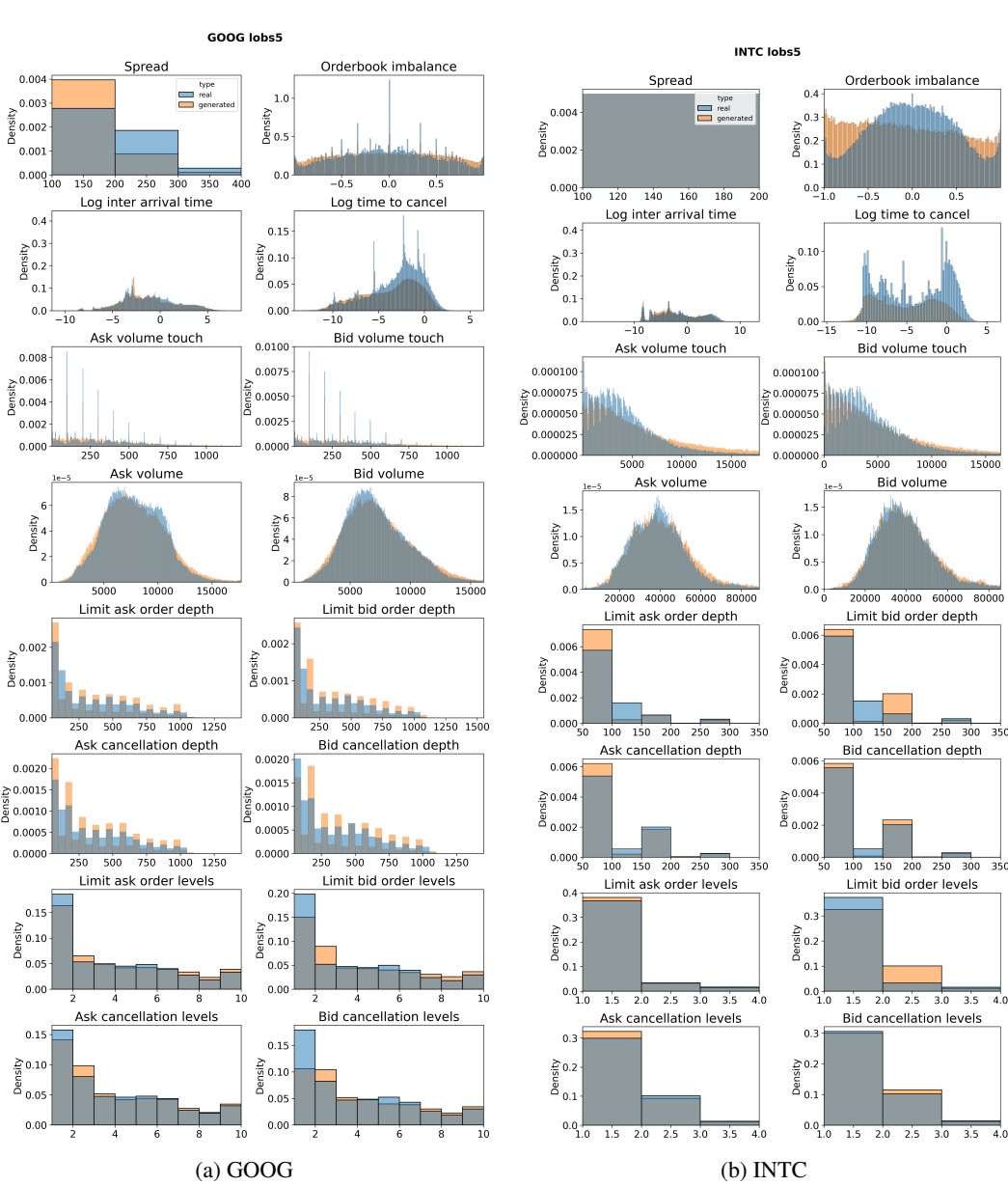

(a) GOOG

(b) INTC

Figure 9: *LOBS5* - histograms comparing score distributions for real (blue) and generated (orange) LOB data for Alphabet (GOOG) and Intel (INTC) stocks. Overall, the generative *LOBS5* model evaluated here, adapted from Nagy et al. (2023), does a good job in matching data along various dimensions. Bigger errors in matching distributions are visible in e.g. spread (GOOG), orderbook imbalance (INTC) and time to cancel (GOOG and INTC).

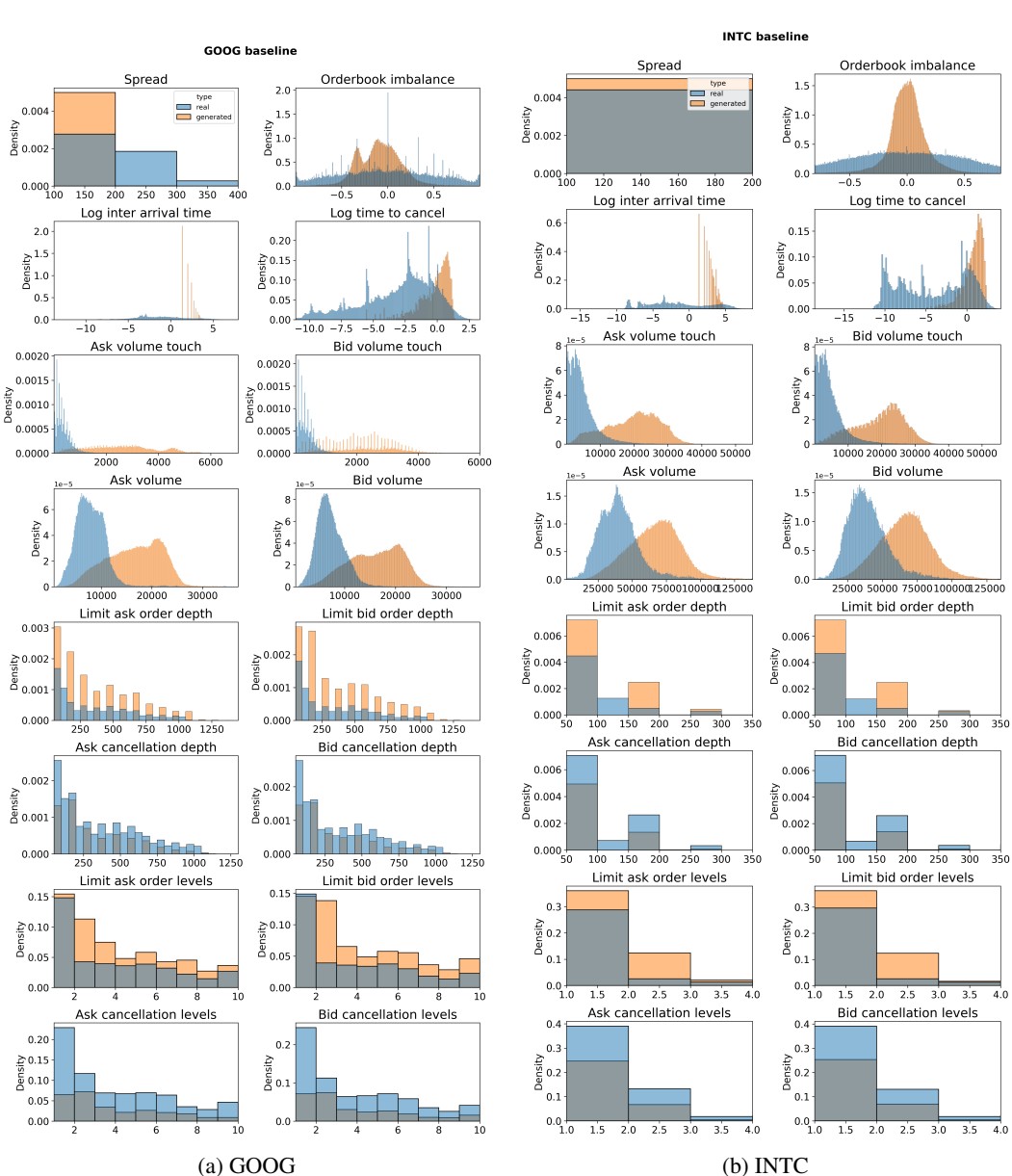

(a) GOOG

(b) INTC

Figure 10: *baseline* - histograms comparing score distributions for real (blue) and generated (orange) LOB data for Alphabet (GOOG) and Intel (INTC) stocks. The Cont et al. (2010) model does a decent job matching some of the scores, particularly discrete ones, such as depths and levels. Clear shortcomings are visible in scores such as orderbook imbalance or volumes.

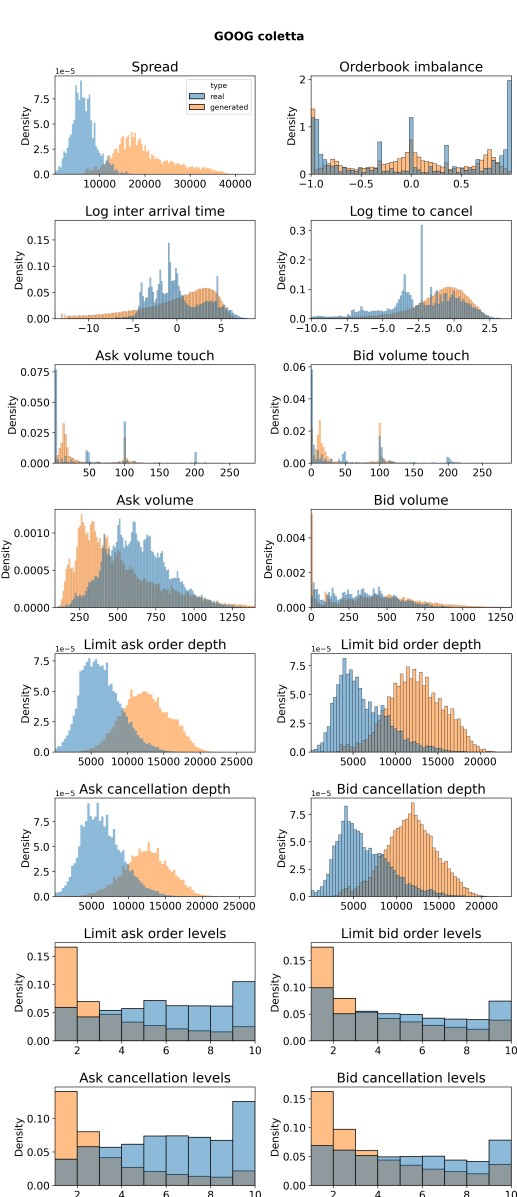

Figure 11: *coletta* - GOOG - histograms comparing score distributions for real (blue) and generated (orange) LOB data for Alphabet (GOOG) and Intel (INTC) stocks.

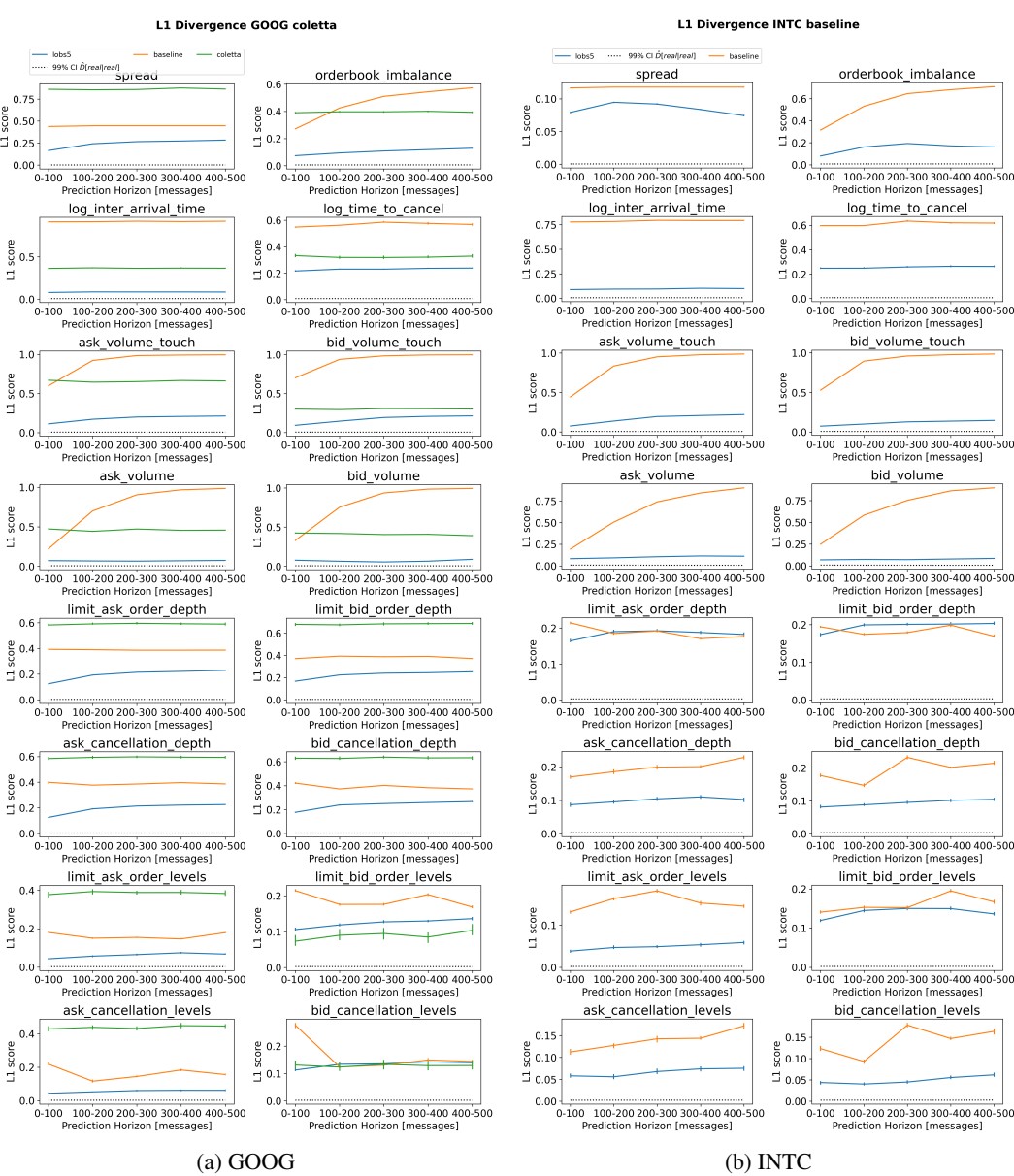

(a) GOOG

(b) INTC

Figure 12: L1 error divergence: comparing the L1 errors of score distributions of real data with generated data distributions at a specific horizon into the future shows accumulating model errors. This is explainable due to snowballing errors caused by teacher forcing (conditional next token loss). A good model should be able to control errors for sequence lengths as long as possible. To provide a significance threshold over pure sampling noise, the dotted lines plot the 99. percentile of L1 error between bootstrapped samples of only real data.

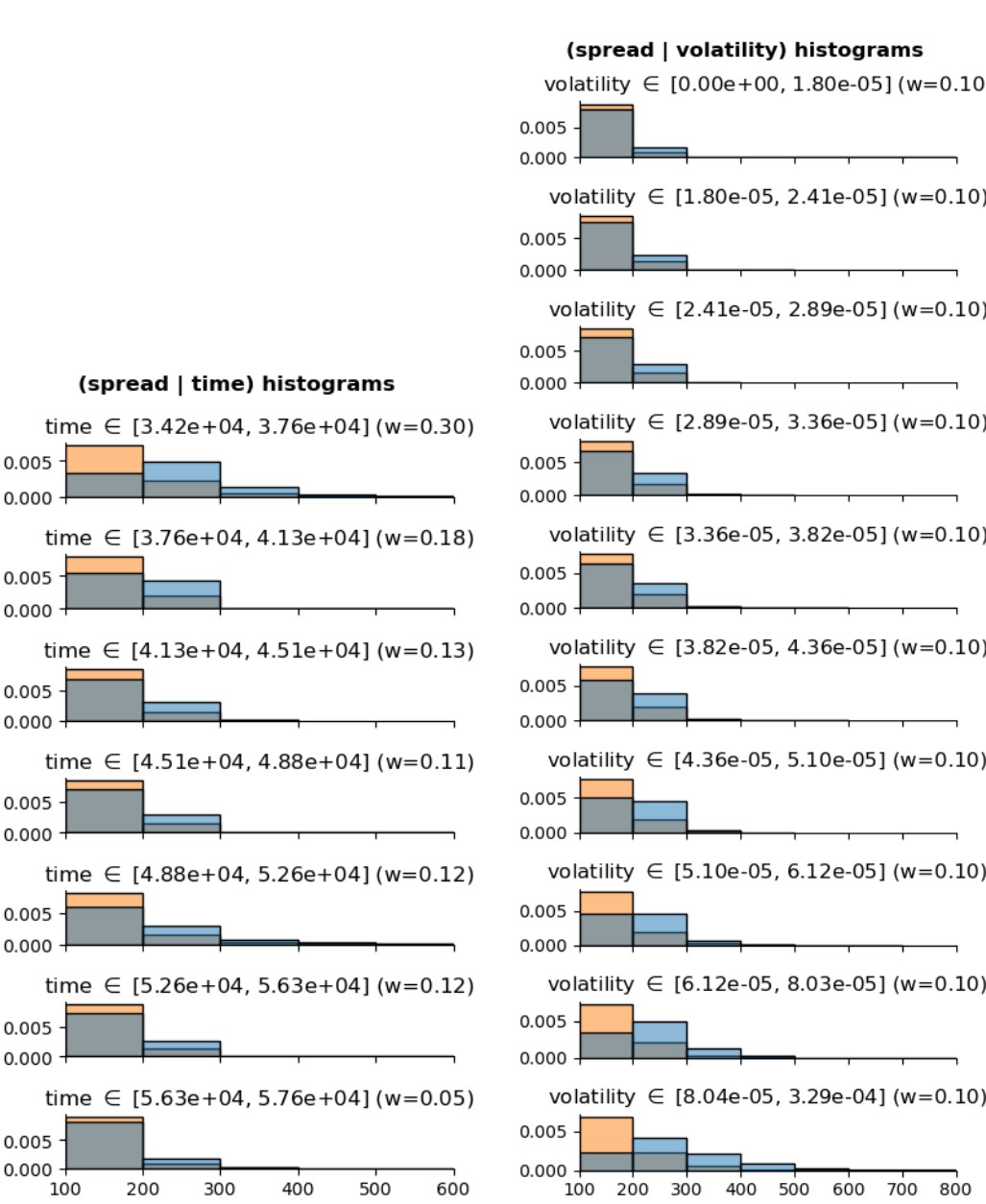

(a) Bid-ask spread conditional on the hour of the day: spreads are higher early in the day, where the generated data also exhibits too narrow spreads.

(b) Spread conditional on volatility: higher volatility corresponds to higher frequency of higher spreads. The model does not fully capture this change, as the higher discrepancy in high-volatility bins shows.

Figure 13: Histograms of conditional score distributions for real (blue) and generated (orange) data for the Alphabet stock (GOOG). Weights $w$, expressing the share of data in the bin, measure the impact of the specific conditional distribution (row) on the total metric loss.

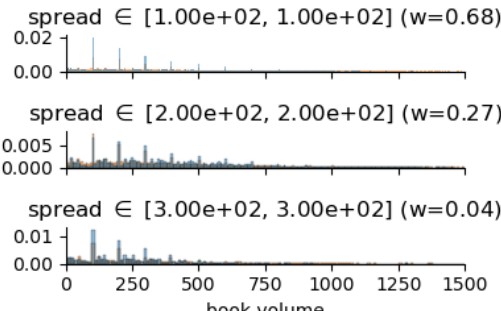

Figure 14: Histograms of total book volume conditional on bid-ask spread for Alphabet stock (GOOG). Weights $w$, expressing the share of data in the bin, measure the impact of the specific conditional distribution (row) on the total metric loss.

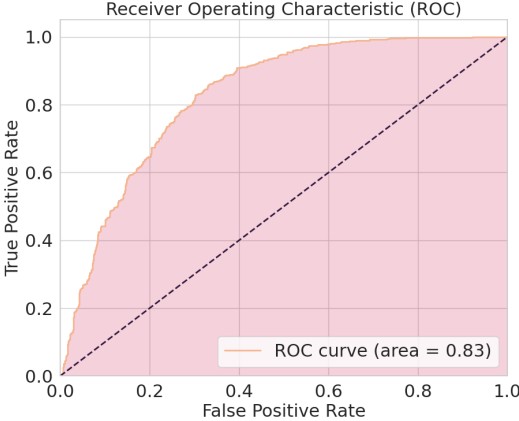

Figure 15: LOBS5 - ROC curve of the discriminator on test data (GOOG). The discriminator represents a worst-case adversarial score function by learning to effectively differentiate between real and generated sequences of LOB states.

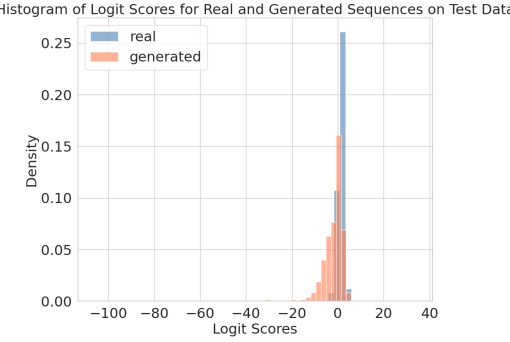

Figure 16: LOBS5 - Histogram of logit scores for real and generated sequences on held-out test data (GOOG). Matching this distribution well would indicate high model quality, as even a trained discriminator network would not be able to differentiate the distributions.

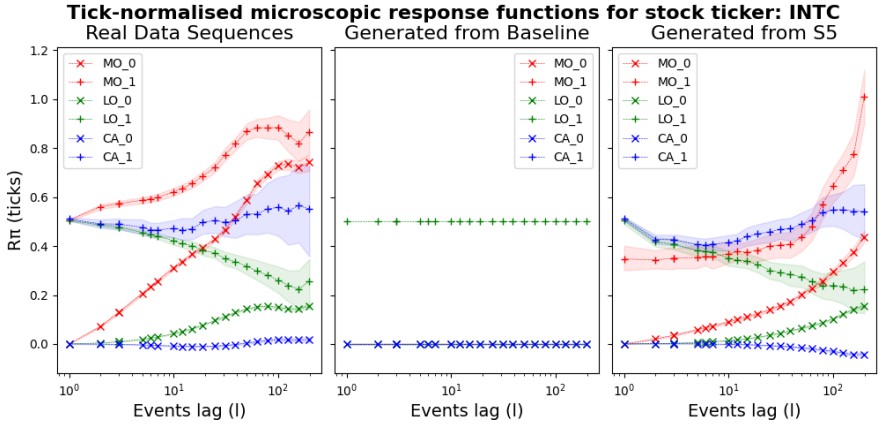

Figure 17: Comparison of impact response functions for different event types between real and generated data-sets, tick-normalized mid-price response. Shaded regions are 99% confidence intervals. There is a comparison between two select models: the LOBS5 and the stochastic baseline. We see that, in contrast to the baseline, the generative model is able to reproduce much more of the expected impact function, though not as well as for GOOG.

