# OpenReview forum: "LOB-Bench: Benchmarking Generative AI for Finance - with an Application to Limit Order Book Markets"
_ICLR.cc/2025/Conference — Submitted to ICLR 2025_

### Official Review · Reviewer_ZXuJ · 2024-10-29

**Soundness:** 2
**Presentation:** 2
**Contribution:** 1
**Rating:** 3
**Confidence:** 4

**Summary:**

The gist of the paper is to evaluate the realism of generated LOB orders. In more detail, the authors want to numerically assess the quality of the generated data instead of just using stylized facts, classic LOB evaluation format, nor only cross-entropy, usually used in generative approaches. The authors want to evalute pre-trained models in *sampling regime*.

**Strengths:**

The paper presents some interesting ideas about evaluating autoregressive models by binning the generated order into some aggregation function $\Phi$ to avoid accumulating errors in time. The quantitative evaluation metric is the L1 distance between $p(x)$ and $\hat{p}(x)$. What this means is still a mystery to me and I'd appreciate to have more clarification from the authors here.

**Weaknesses:**

### Discussion over the meaning of L1 as a qualitative evaluation metric
I appreciate the authors taking the effort of running all stylized facts experiments and stress-testing 2 SoTA and 1 baseline method. However, the authors claimed in the beginning that just having stylized facts, already thoroughly explored by Vitrienko et al. [5], isn't sufficient. While they support this claim by introducing the L1 as a qualitative metric, it lacks substance and overpresenting these stylized facts overshadows the "novelty" of the benchmark. Anyhow, to the best of my understanding, the L1 should quantitatively say how good a LOB generation model performs. Nevertheless, I argue that it'd be simpler and more than enough to train a model -- even a simple one -- on a financial downstream task -- say mid-price prediction -- on "fake" generated orders and measure its MSE. Then, at test time, one would measure the model's MSE on real orders and see how well the model trained on generated orders generalizes over the real data. This would be a sufficient indicator to assess the goodness and realism of the generated data. Furthermore, you wouldn't even need to bin your data and go through unnecessary hoops just for the sake of the benchmark. So, the question raised here is *Why is the literature in LOB generation "dying" to have your contribution? What are you bringing to the table that future works are going to benefit from?*

### Shortsightness and lack of discussion within the benchmark
As a benchmark paper, I would expect the authors to give insights and directions for future autoregressive/generative LOB models. This is lacking which undermines its usefulness. Again *What is it that future researchers would benefit from this benchmark?*

### Weaknesses
1. **Why did you share your GitHub repository not anonymized? Now, I know your identity and have reported it to the ACs**.
2. Lines 69-84 represent the authors' contribution and the writing is sloppy and all over the place. So are the authors trying to evaluate how good SoTA methods predict the mean-price on the newly generated orders? Doesn't this just cover the cross-entropy aspect that GenAI literature uses for evaluation? Where's the realism here?
3. It is unclear from lines 86-89 why model derailment is due to true data seeding. Care to discuss?
4. There's missing information about *coletta* and limited results for GOOG (lines 390-394). What missing info?
5. In Figure 6, it is evident that the trend of MO$_0$, and, especially, MO$_1$ aren't captured by *lobs5*. How do you justify this?

Minors:
1. There might be a missing reference in line 76 where you stat "Our aggregator functions are closely inspired by metrics used in literature,
such as spread etc.". How are they inspired by the literature? Which work specifically are you inspired from?
2. Who said that GANs are worst-case aggregators? Why is that?
3. It would be better to introduce already the models you're using in the benchmark in line 86. Why only 2 models if you're proposing a benchmark? There are plenty of other generic time series generation models that you can try to adapt for LOB [1-4].
4. Reference sections/equations correctly:
* e.g., line 265 $\rightarrow$ Equation~\eqref
* e.g., line 307 $\rightarrow$ Section~\ref
5. Lines 97-98 and 98-99 are the same sentence. This is where the authors also reveal their identity.
6. Nobody calls cross-entropy *xent*. Please amend this.


[1] Lim H, Kim M, Park S, Park N. Regular time-series generation using SGM. arXiv preprint arXiv:2301.08518. 2023 Jan 20.

[2] Li J, Wang X, Lin Y, Sinha A, Wellman M. Generating realistic stock market order streams. In Proceedings of the AAAI Conference on Artificial Intelligence 2020 Apr 3 (Vol. 34, No. 01, pp. 727-734).

[3] Shi Z, Cartlidge J. Neural Stochastic Agent-Based Limit Order Book Simulation: A Hybrid Methodology. In Proceedings of the 2023 International Conference on Autonomous Agents and Multiagent Systems 2023 May 30 (pp. 2481-2483).

[4] Hultin H, Hult H, Proutiere A, Samama S, Tarighati A. A generative model of a limit order book using recurrent neural networks. Quantitative Finance. 2023 Jun 3;23(6):931-58.

[5] Vyetrenko S, Byrd D, Petosa N, Mahfouz M, Dervovic D, Veloso M, Balch T. Get real: Realism metrics for robust limit order book market simulations. In Proceedings of the First ACM International Conference on AI in Finance 2020 Oct 15 (pp. 1-8).

## **Even in case the authors didn't reveal their identity, I would still have rejected this paper**

**Questions:**

See weaknesses, and the following:

1. Have you tried the KL-divergence for $\mathbb{D}$ in Eq. 1? Why is L1 more suitable? Can you run some experiments please to see this effect?
2. Does Figure 2 measure the L1 loss on all these dimensions?
3. Fig 15 $\rightarrow$ So, you trian a GAN on real and generated data from *lobs5* and you achieve an AUCROC of 82\%? This means that the generated orders are easily identified as fake, undermining what the authors claim in their original works. How come? What's the AUCROC for *coletta* and the *baseline*?
4. I might have missed it but what's the *baseline* model here?

**Details Of Ethics Concerns:**

* **Responsible research practice (data release)**: the used GOOG stock in the paper is not freely available undermining the authors' claims and the reproducibility of their results

* **Other reasons (anonymity violation)**: The authors provided, in the end of the introduction, a "clean link" to their GitHub repository where I can pinpoint the authors' affiliation and their identity.

---

> ### Author Response · Authors · 2024-11-22
> **Replies to Reviewer ZXuJ**
>
> **Clarifications**
>
> We believe there may have been a misunderstanding regarding the central scoring methodology used by LOB-Bench. To clear this up, we added additional discussion on the methodology, a new Figure 1 to illustrate the process and explicit paper contributions in the introduction.
>
> **Replies to Concerns**
>
> > just having stylized facts, already thoroughly explored by Vitrienko et al. [5], isn't sufficient. While they support this claim by introducing the L1 as a qualitative metric, it lacks substance and overpresenting these stylized facts overshadows the "novelty" of the benchmark.
>
> The point of a benchmark is to provide a theoretical framework of how to measure generated data quality, as well as, a useful practical tool for further model development. The statistics explored in Vyetrenko et al. are amongst those relevant for financial practitioners and users of such models, and are therefore necessary elements of a good benchmark. In our paper, we also show that all such statistics neatly fit into the theoretical framework of scoring functions as dimensionality reduction operators. Because models in this domain have so far only managed to qualitatively match "stylized facts", we argue that further model development is currently limited by exactly quantifying *how* well the model matches the actual data distribution, rather than only saying qualitatively that a model can reproduce some expected behaviour or some emerging characteristic has the expected magnitude. And that we provide by quantifying distributional similarity, e.g. by using an L1 distance.
>
> Furthermore, we show that our framework is general enough to also encompass adversarial scoring functions, derived from GAN training. So, while statistics of interest to practical applications are essential, they are not of sole relevance.
>
> Besides our theoretical contribution, we would argue that a benchmark's value mainly lies in its usefulness as a practical tool and ability to provide a measuring stick to further future model development.
>
> > I argue that it'd be simpler and more than enough to train a model -- even a simple one -- on a financial downstream task -- say mid-price prediction -- on "fake" generated orders and measure its MSE. Then, at test time, one would measure the model's MSE on real orders and see how well the model trained on generated orders generalizes over the real data. This would be a sufficient indicator to assess the goodness and realism of the generated data.
>
> This is an interesting suggestion, which we had also considered. However, there are a number of issues, which unfortunately make this approach impractical.
>
> * A significant sim-to-real transfer gap can be expected even for very good generative models, if the "simple" model trained to perform the task is *too simple*. Many financial applications, such as forecasting are notoriously hard to do well, requiring a lot of development work, which makes such a benchmark complex to perform.
> * Even using a *strong model*, trained on *good generative* data will have some transfer performance loss as the model cannot be perfect. So comparing like-for-like would require comparing real-to-real with sim-to-real transfer performance, requiring two models to be trained for each task, making such a benchmark even more complex.
> * More damagingly, any concrete downstream task can only evaluate a very narrow range of a model's abilities. Mid-price prediction, for example, could never quantify a model's ability to accurately model behaviour deeper in the book, which could be very important for other applications. Furthermore, using MSE as an evaluation metric wouldn't even measure very simple distributional characteristics, such as variance, and would therefore be a bad choice. Such an approach would therefore never be a sufficient indicator to assess realism.
> * For many current generative LOB models, sim-to-real transfer would simply fail because the model performance is not strong enough yet. Whether a model can perform a concrete task or not can also be a very discrete difference. Such failure would however be very hard to quantify, and therefore not be as useful in a benchmark, which is aimed at gradually improving model performance.
>
> For these reasons, we argue that our approach of measuring distributional differences is the clear better choice to accelerate model development at the current stage of research.

---

> > ### Author Response · Authors · 2024-11-22
> > **Answer to Questions (Reviewer ZXuJ)**
> >
> > > As a benchmark paper, I would expect the authors to give insights and directions for future autoregressive/generative LOB models. [...] Why is the literature in LOB generation "dying" to have your contribution? What are you bringing to the table that future works are going to benefit from?
> >
> > We have now clarified in the paper that LOB-Bench is the first benchmark of its kind, by evaluating fully generative models of limit order book data. Benchmarks so far had a severly limited scope, e.g. by focusing only on mid-price forecasting. Our **contributions** are as follows:
> >
> > * We provide the first LOB benchmark to address full *distributional* quantification of model performance. Prior work relied on matching qualitative descriptions of *stylized facts*, making model comparisons impossible and stymying development.
> > * Interpretable scoring functions of key interest to practitioners allow targeted model development and improvement.
> > * Discriminator scores provide a difficult benchmark to beat for future generative models, even if most other statistics were to match closely.
> > * Divergence metrics (distributions as functions of unroll steps) highlight a common failure mode, which can now be addressed in future model generations.
> > * The benchmark is easy to use and only requires data in the common LOBSTER format.
> > * LOB-Bench is easily extensible to additional scoring functions.
> > * The theoretical framework is transferable to other generative time-serie domains with high-dimensional data.
> >
> > Ultimately, we believe that LOB-Bench exhibits the key criteria required for a good benchmark, by providing the measuring stick along which growth in the field can unfold.
> >
> > **Minor Points**
> >
> > > There might be a missing reference in line 76 where you stat "Our aggregator functions are closely inspired by metrics used in literature, such as spread etc.". How are they inspired by the literature? Which work specifically are you inspired from?
> >
> > Thank you for the comment. We have added citations substantiating this.
> >
> > > Who said that GANs are worst-case aggregators? Why is that?
> >
> > The discriminator aims to find the "worst-case" function $\Phi^*$ that maximally separates the real and generated distributions by choosing $\Phi^*$ such that it maximizes the divergence between them, i.e., $\Phi^* = \arg\max_{\Phi} D[p(\Phi(d)), \hat{p}(\Phi(d)))]$.
> >
> > > It would be better to introduce already the models you're using in the benchmark in line 86.
> >
> > We have now added this at this position as well.
> >
> > > Reference sections/equations correctly
> >
> > Thank you for pointing this out -- this has now been fixed.
> >
> > > Nobody calls cross-entropy xent. Please amend this.
> >
> > We have changed instances of "xent" to "cross-entropy".
> >
> > **Replies to Questions**
> >
> > > **Q1** Have you tried the KL-divergence for D in Eq. 1? Why is L1 more suitable? Can you run some experiments please to see this effect?
> >
> > We have added a justification for the use of L1 and Wasserstein metrics to the paper.
> >
> > The KL divergence tends to be unstable for some empirical distributions. When the data is binned in a way that one distribution has no observations in a single bin, the KL is infite, and hence tends to explode in many cases. Good estimates of KL would require intermediate steps of continuous density estimation and make the pipeline less robust and more complex. Furthermore, L1 and Wasserstein are *metrics*  by satisfying commutativity, and have a more intuitive interpretation.
> >
> > > **Q2** Does Figure 2 measure the L1 loss on all these dimensions?
> >
> > The model score summaries include all conditional and unconditional scoring metrics. Market impact scores, and divergence errors are reported separately.

---

> > > ### Author Response · Authors · 2024-11-22
> > > **Reply to questions and Ethics Concerns (Reviewer ZXuJ)**
> > >
> > > > **Q3** So, you tr[ai]n a GAN on real and generated data from lobs5 and you achieve an AUCROC of 82%? This means that the generated orders are easily identified as fake, undermining what the authors claim in their original works. How come? What's the AUCROC for coletta and the baseline?
> > >
> > > We achieved an AUC-ROC score of 0.82 for the autoregressive model evaluated with a conv-transformer discriminator architecture. For contrast, for both other models, the test set AUC-ROC is almost 1, indicating that these models fail completely in the discriminator scoring function. We have clarified these results in the paper.
> > >
> > > We don't train a GAN but only a discriminator, which acts as a scoring function. A high AUCROC indeed indicates that the generated data is easily differentiable from real data. This could in principle be due to artifacts in the generated data, which do not have an immediate effect on downstream model performance, but may more likely indicate model failure. Bad discriminator metrics (i.e. high L1 of resulting score distribution) could point a way to potential model improvement by e.g. including GAN model finetuning or other sequence level error reduction methods.
> > >
> > > LOB messages are a challenging domain as real data is generated from a complex system of a large number of intelligent agents. Even though generative models have shown much progress recently, there is still a significant gap towards hyper-realistic large financial models, which even good discriminator networks cannot differentiate from real data. Furthermore, the discriminator works on the sequence level, not on individual messages, making it easier to distinguish real from fake sequences. Hence, being able to differentiate such generated data from real data is the expected result, even for SotA models.
> > >
> > > > **Q4** I might have missed it but what's the baseline model here?
> > >
> > > The baseline model is the model from Cont et al. (2010) [2], as outlined at the start of section 6. We have now also already introduced the models under consideration at the end of the introduction.
> > >
> > > **Ethics Concerns**
> > >
> > > > The used GOOG stock in the paper is not freely available undermining the authors' claims and the reproducibility of their results
> > >
> > > We respecfully disagree that this constitutes an ethics violation. While the LOBSTER dataset is not freely available, it is available for a low cost to academic institutions and is the de-facto standard for academic research in this domain. There is also a free data sample available on their website, which allows experimentation with the benchmark package. We would ourselves prefer a completely free dataset, however such a dataset, which is sufficiently large to train modern ML models, is not available. Synthetically generated data might be able to fill this gap in the future, once models are sufficiently accurate. To get there, benchmarks such as LOB-Bench are key accelerators.
> > >
> > > We will publish a sample of generated data online before the paper is published.
> > >
> > > > The authors provided, in the end of the introduction, a "clean link" to their GitHub repository where I can pinpoint the authors' affiliation and their identity.
> > >
> > > As already mentioned, this link was left in by mistake. We apologise for the oversight. This has been fixed.
> > >
> > > ## Conclusion
> > >
> > > We hope that we could address most of your questions and concerns and that the changes to the paper merit an improved score. We would equally welcome suggestions on how to improve the paper further.
> > >
> > > ## Literature
> > >
> > > [1] Nagy, Peer, et al. "Generative ai for end-to-end limit order book modelling: A token-level autoregressive generative model of message flow using a deep state space network." Proceedings of the Fourth ACM International Conference on AI in Finance. (2023).
> > >
> > > [2] Cont, Rama, Sasha Stoikov, and Rishi Talreja. "A stochastic model for order book dynamics." Operations research 58, no. 3 (2010): 549-563.

---

> > ### Comment · Reviewer_ZXuJ · 2024-11-26
> > **MSE sim-to-real discussion**
> >
> > Ok that you have these concerns, but trying it would be worthwhile. Regarding your second point (i.e., comparing like-for-like and the complexity of the benchmark) doesn't mean that it's complex. It's just how you need to evaluate this kind of downstream model. First you train the model on the real data and test it on the market replay after a certain period. You measure the MSE here and save it as the baseline metric. Then you use an order generator and trian the same model (from scratch) on it and test it on the market replay. You then compare these MSEs which allows you to effectively know whether the generated orders are "real enough", which goes against your statement about realism. I.e., if the MSE here is low and approaches the MSE of the baseline, the generation process is good.
> >
> > I believe that your arguments here are vague since there's already some works in finance that adopt this "predictive score" to assess the goodness of an order generator (see [1] for example).
> >
> > [1] Yoon J, Jarrett D, Van der Schaar M. Time-series generative adversarial networks. Advances in neural information processing systems. 2019;32.

---

> > > ### Author Response · Authors · 2024-11-27
> > > **MSE sim-to-real discussion**
> > >
> > > Besides the complexity argument, we also explicitly formulated 3 other reasons why we did not use this sim-to-real framework as part of our benchmark - none of these are vague:
> > > * difficulty of developing a good model for mid-price prediction,
> > > * only narrow evaluation by using a single task,
> > > * predictive failure provides a limited signal for improvement.
> > >
> > > In the framework of Yoon et al. the *predictive score* makes much more sense than for LOB message generation because it is possible to forecast the generated data using an MSE loss. In our case of LOB messages, that would entail predicting all 6 message features in LOB, including complexities such as limit prices, levels, sizes, order types, referencing previous orders for cancellations etc. An MSE loss on this data makes no sense.
> > > Reducing the task to something simple, where MSE would make send, e.g. mid-price forecasting is a very limited mode of evaluation as there are many other characteristics of LOBs, which are not captured by mid-price behaviour, e.g. separate bid/ask behaviour, traded volume, available volume at different levels etc. In contrast, our distributional evaluation allows us to cover all of these with the purpose of developing a *general-purpose* "Large LOB Model".
> > > So, while sim-to-real transfer on mid-price prediction *does* measure *something* about the model performance, this is orthogonal to the distributional framework we use in LOB-Bench.
> > >
> > > Overall, we do not follow the argument that our framework is bad because someone else has done it differently. This goes exactly against the purpose of doing novel research.

---

> ### Comment · Reviewer_ZXuJ · 2024-11-26
> **More stuff**
>
> So what is the AUC-ROC for Coletta and the baseline? You still didn't give me an answer here.
>
> Still using LOBSTER data can't be accessible by all research institutions. In fact, all papers in financial time series and LOB market simulations rely on third-party data which makes these papers not reproducible. What about releasing a LOB dataset that comes for free with the benchmark, like the FI-2010 dataset but with the message book? That would be something that would actually **accelerate the way to better generation models**.
>
> Yeah, just saying sorry about having missed the clean link in your paper doesn't smooth the breach of anonymity. I'm upping my score to 3 (reject, not good enough), but will still argue to get it rejected.

---

> > ### Author Response · Authors · 2024-11-27
> > **More stuff - reply**
> >
> > For both these models the AUC-ROC is 1.0, so our discriminator model can perfectly identify fake sequences.
> >
> > Unfortunately, we are bound by the license agreements in that we cannot just publish LOBSTER data. What we can do—and intend to do—is publish a generated data sample that researchers could use for experiments. However, such data has obvious weaknesses, and we know that this cannot replace real data for many applications yet.

---

> ### Public Comment · ~Leonardo_Berti2 · 2025-03-01
> **In my paper (TRADES: Generating Realistic Market Simulations with Diffusion Models) I did what the reviewer is suggesting, I have calculated the predictive score for generated data, using 2 of the same models that LOB-Bench uses.**
>
> In my paper (TRADES: Generating Realistic Market Simulations with Diffusion Models), I did what the reviewer suggested. I have calculated the predictive score for generated data, using 2 of the same models that LOB-Bench uses. Just to be clear, I did this at the end of 2023 (proof commit date on my GitHub), but I decided to publish the paper only 2 months ago.

---

### Official Review · Reviewer_trTG · 2024-11-02

**Soundness:** 2
**Presentation:** 2
**Contribution:** 2
**Rating:** 1
**Confidence:** 3

**Summary:**

This paper proposes a unified benchmark framework, LOB-Bench, designed to quantitatively evaluate generative AI models with a focus on limit order book (LOB) data. Unlike prevailing qualitative measurements, this framework provides numerical results to analyze the accuracy of how the LOB generative models resemble the real data. Compared to current quantitative metrics in the literature, such as cross-entropy, LOB-Bench is claimed to better deal with issues like distribution shifts and autoregressive gaps within data. LOB-Bench is applied to evaluate a range of generative models. The experiment section presents some interesting findings. Specifically, the results indicate that autoregressive GenAI’s excel in learning LOB data more effectively than traditional methods.

**Strengths:**

1. This paper highlights the essential demands for quantitative metrics to evaluate generative AI for finance and addresses this critical problem by introducing several innovative quantitative metrics, which are well-motivated and useful for the further development of generative AI in finance.
2. This paper effectively employs visualizations to show the quantitative results, making them intuitive and comprehensive for a broad audience.

**Weaknesses:**

1. This paper provides a non-anonymous code link on line 097, which violates the double-blind protocol of the conference. The authors should take care to remove or anonymize the link for the review process.
2. This paper displays several signs of incomplete editing that require further proofreading. Specific issues include a non-functional GitHub link on line 033, and repeated sentences between lines 097-099. Before submitting, the authors should conduct a thorough proofreading pass, paying particular attention to consistency in links and removing any redundant text.
3. Although the paper introduces a new benchmark for evaluating generative AI models, it does not adequately compare this benchmark with existing alternatives. Furthermore, the experiments fail to convincingly demonstrate the benchmark's claimed ability to effectively address distribution shifts or autoregressive gaps within the generated data. What are some existing benchmarks against which the proposed method can be compared against? Specific experiments that clearly demonstrate superiority in dealing with distribution shift or autoregressive gaps in the generated data would make the method more convincing.

**Questions:**

1. As mentioned above, this paper does not demonstrate that LOB-Bench addresses autoregressive gaps and distribution shifts effectively. Can you provide a detailed analysis or case studies that illustrate the benchmark's performance in managing these specific challenges?
2. Could you demonstrate how LOB-Bench differentiates between generated and real LOB data in certain scenarios where traditional methods such as cross-entropy fail to?
3. How does the proposed benchmark influence the development and refinement of generative AI models? Are there examples where insights derived from LOB-Bench can directly lead to improvements in model architecture or training methods?

---

> ### Author Response · Authors · 2024-11-22
> **Replies to Reviewer trTG**
>
> **Clarifications**
>
> > Compared to current quantitative metrics in the literature, such as cross-entropy, LOB-Bench is claimed to better deal with issues like distribution shifts and autoregressive gaps within data.
>
> Our goal for this benchmark is to be agnostic to the model class generating the data. One consequential limitation is that we cannot use cross-entropy, as not all model classes generate explicit distributions over tokens, or the generated data sequence more generally. For example, agent-based simulations or CGAN models may only generate a single data realisation for a given random seed, rendering cross-entropy calculations infeasible. We have added this point to the paper.
>
> A key contribution of LOB-Bench is the evaluation of generated distributions, conditional on the unrolling step (see p. 17, Figure 11). This allows an evaluation of distribution shifts due to autoregressive error divergence.
>
> **Replies to Concerns**
>
> > This paper displays several signs of incomplete editing that require further proofreading. Specific issues include a non-functional GitHub link on line 033, and repeated sentences between lines 097-099. Before submitting, the authors should conduct a thorough proofreading pass, paying particular attention to consistency in links and removing any redundant text.
>
> We thank reviewer **trTG** for pointing out editing mistakes that slipped through the editing process, which have now been fixed in the updated manuscript.
>
> > Although the paper introduces a new benchmark for evaluating generative AI models, it does not adequately compare this benchmark with existing alternatives.
>
> A major reason for creating this benchmark was that no comparable benchmark for generative LOB data exists yet. But to provide some more general context, we have added a discussion of other benchmark methodologies for financial data to the paper.
>
> > Furthermore, the experiments fail to convincingly demonstrate the benchmark's claimed ability to effectively address distribution shifts or autoregressive gaps within the generated data. [...] Specific experiments that clearly demonstrate superiority in dealing with distribution shift or autoregressive gaps in the generated data would make the method more convincing.
>
> :question: Could you kindly clarify what kind of distribution shift you think we are not addressing adequately?
>
> We are addressing two types of distribution shift directly in the benchmark: conditional distribution shift and autoregressive distribution shift. By allowing the evaluation of arbitrary conditional distributions, the benchmark can evaluate how a distribution of interest shifts, conditional on another scoring function. For example, $p(\text{spread} | \text{volatility})$ allows evaluating how well the model is able to learn this type of distribution shift of the spread. Even more generically, $p(\text{spread} | \text{time})$ captures distribution shift of the spread by time of day (see p. 18, Figure 12).
>
> The second kind of distribution shift captured by the benchmark is autoregressive distribution shift during inference. The benchmark measures how quickly generated distributions shift from the target distribution, supporting future model development to mitigate this type of failure mode.
>
> Since we unfortunately cannot provide raw data with the benchmark due to licensing limitations, the selection of data to use with the benchmark is up to the user, since we also do not know which data their model was trained on. To evaluate a distribution shift between train and test, users could e.g. train their models on data before 2020 and evaluate during 2021, when market behaviour may have differed due to the Covid turbulence.
>
> We have clarified our discussion of these points in the paper.

---

> > ### Author Response · Authors · 2024-11-22
> > **Replies to Reviewer trTG's Questions**
> >
> > **Replies to Questions**
> >
> > > **Q1** As mentioned above, this paper does not demonstrate that LOB-Bench addresses autoregressive gaps and distribution shifts effectively. Can you provide a detailed analysis or case studies that illustrate the benchmark's performance in managing these specific challenges?
> >
> > We have argued above that the benchmark *does* address different kinds of distribution shift, by conditioning distributions on other variables (e.g. volatility, volumes, time, ...), by evaluating autoregressive error divergence, and also by allowing different train and test time periods, which may be chosen to exhibit distribution shift. The main point is that by measuring distributional differences, all these types of distribution shifts come naturally to the benchmark.
> >
> > :question: We would welcome any further suggestions of what other changes or experiments would help to even better address distribution shift.
> >
> > > **Q2** Could you demonstrate how LOB-Bench differentiates between generated and real LOB data in certain scenarios where traditional methods such as cross-entropy fail to?
> >
> > As explained above, unfortunately, cross-entropy is not possible to calculate for all model classes (e.g CGAN, agent-based models), making this type of evaluation infeasible for a general-purpose, model-agnostic benchmark. For those model classes admitting explicit distributions, such as autoregressive models, of which we are only aware of a single model [1], cross-entropy would usually already be the training loss, and hence not add much new information in the form of a benchmark.
> >
> > Even if possible, cross-entropy based evaluation, e.g. by calculating test data *perplexity*, does not provide detailed information about specific failure cases and areas of model strengths and weaknesses. To judge the usefullness of a model, we should care about the emergent aggregate behaviour of the generative model. To make an anology with LLMs, development is largely driven by a battery of benchmarks, measuring skills beyond just next word prediction, which is an imprecise measure of the quality of the model. We want to provide a similar tool for generative LOB models.
> >
> > > **Q3** How does the proposed benchmark influence the development and refinement of generative AI models? Are there examples where insights derived from LOB-Bench can directly lead to improvements in model architecture or training methods?
> >
> > By measuring a number of different distributions, LOB-Bench allows evaluating generative models along different dimensions. As demonstrated in the paper, distributions of interest can be calculated using *interpretable* scoring functions, e.g. spread of message inter-arrival times, or more *abstract* scoring functions, e.g. discriminator scores. This highlights specific model strengths and weaknesses and helps researchers develop solutions to fix problems identified by the benchmark. Another direction of improvement for SotA models is highlighted by the autoregressive divergence problem, which LOB-Bench evaluates by measuring distributions conditional on the unroll step.
> >
> > We have now clarified in the manuscript the impact we think LOB-Bench will have on the field.

---

### Official Review · Reviewer_GuKS · 2024-11-04

**Soundness:** 3
**Presentation:** 3
**Contribution:** 3
**Rating:** 6
**Confidence:** 4

**Summary:**

The paper presents the LOB-Bench, a benchmark designed to evaluate the quality and realism of generative models for Limit Oder Book(LOB) data. It provides a theoretical framework and Python package that systematically compares generative models like autoregressive models, (C)gAN, and agent-based models. It uses a range of metrics like spread, order book volumes, order imbalance, and adversarial scores to asses the distributional differences between generated and real LOB data. Empirical results demonstrate that the autoregressive generative approach outperforms traditional models, offering a standardized evaluation for financial data generation.

**Strengths:**

1) The paper proposes unconditional and conditional evaluations, allowing for a nuanced comparison of generative models. It accounts for model drift and evaluates market impact metrics, making the framework robust.

2) The use of widely adopted LOBSTER data for empirical evaluations makes the benchmark relevant for practitioners in the financial domain.

3) The open-source code enhances the reproducibility and practical unity of the work.

**Weaknesses:**

1) The paper explores generative models; some more, like advanced deep learning architectures, could be explored.
2)  The evaluation highlights that errors accumulate over longer sequences. The benchmark does not fully address how to mitigate this.
3) Certain model results with rate events like market orders have data sparsity. and generating additional data might not be practical.

**Questions:**

1) The framework does not account for hidden liquidity, which is critical for LOB modeling. The model can be improved in this area.
2) Why did you focus on autoregressive models, (C)GANs, and agent-based models? Have you considered other deep learning architectures for LOB data generation, such as transformers or diffusion models?
3) Could you clarify why you selected the L1 and Wasserstein-1 distances as the primary metrics for distributional comparison? How do these metrics handle the complexities of financial time series data, such as fat-tailed distributions or volatility clustering?
4) The benchmark evaluates market impact metrics, but how do you account for outliers or extreme events in real-world LOB data? Are these adequately captured in your current framework?
5)  The conditional evaluation approach looks at conditional distributions of specific LOB statistics. How would this framework perform when dealing with very high-dimensional data, and does it scale well for more complex scenarios?
6) The paper uses the LOBSTER dataset for evaluation. Given its limitations (e.g., limited asset coverage, certain missing features), do you think the results would generalize to other datasets or more diverse market conditions?

---

> ### Author Response · Authors · 2024-11-22
> **Replies to Reviewer GuKS**
>
> **Replies to Concerns**
>
> > The paper explores generative models; some more, like advanced deep learning architectures, could be explored.
>
> The GenAI literature for structured financial data, such as LOBs, is still a nascent field. We picked two recent SotA "advanced deep learning architectures" in different classes (CGAN, autoregressive), as well as a more classical statistical model for evaluation. While not a wide variety of advanced deep learning models have been published in this domain, we also do not claim to have performed a comprehensive benchmark evaluation of *all* available models. Instead, our aim is to provide this benchmark as a tool to further aid the development of new models and to provide a unified solution for comparable data formats in the field. Furthermore, comparing two SotA models, we make the case that autoregressive models appear to have an edge in this data modality and suggest this as a promising direction for further research.
>
> > The evaluation highlights that errors accumulate over longer sequences. The benchmark does not fully address how to mitigate this.
>
> We would argue that a benchmark *does in principle* never take the roll of providing solutions how to mitigate types of model errors. Instead, by elucidating specific failure modes, such as error accumulation, the benchmark can aid in developing new models and methods aimed at solving such issues.
>
> In principle, methods reducing errors on a sequence level, or RL finetuning on downstream tasks could be viable options to improve models in this direction.
>
> > Certain model results with ra[r]e events like market orders have data sparsity. and generating additional data might not be practical.
>
> We have not yet run into problems with data sparsity, particularly with market orders. Besides, the LOBSTER data standard does not differentiate between executions due to executable limit orders and market orders. In any case, execution order types are frequent enough for generative models to learn them. Generating more synthetic data is also always possible in principle.
> Moreover, current generative models for LOB data still have problems learning distributions accurately, even when ignoring outliers.
>
>
> **Replies to Questions**
>
> >**Q1** The framework does not account for hidden liquidity, which is critical for LOB modeling. The model can be improved in this area.
>
> We used the same data pre-processing pipeline for the training of all models in the paper. We indeed remove all hidden order types (type 5 in LOBSTER), which denote the execution of an incoming order against a hidden liquidity block at the end of a queue. By removing this altogether, we do not claim to model hidden liquidity but create an equal footing for each model. The GenAI task is hence to model LOB behaviour with the effects of hidden liquidity clearly removed.
>
> :question: If reviewer **GuKS** has concrete examples of cases where omitting hidden liquidity in this way would affect results, we would welcome such suggestions. To the best of our knowledge, there are no models for generating market-by-order data which also attempt to model the hidden liquidity.
>
>
> >**Q2** Why did you focus on autoregressive models, \(C)GANs, and agent-based models? Have you considered other deep learning architectures for LOB data generation, such as transformers or diffusion models?
>
> A key feature of the benchmark is that it is agnostic to the model class, which generates the data. All that is needed for compatibility is that the model generates data in the LOBSTER data format. We evaluate a \(C)GAN and an autoregressive state-space-model, as these are two SotA models in the field, and a more classical statistical model which is commonly used in the finance literature. We further mention agent-based models, as these are also commonly used in the literature and have added  corresponding references to the paper. Developers of transformer based sequence models or diffusion models of LOB dynamics could also evaluate their novel models using LOB-Bench. Generally, LOB data has a structured format and is not usually compatible with out-of-the box sequence models for a various reasons, such as non-stationarity, arbitrary order ids, nanosecond time precision etc.

---

> > ### Author Response · Authors · 2024-11-22
> > **Further replies to questions.**
> >
> > >**Q3** Could you clarify why you selected the L1 and Wasserstein-1 distances as the primary metrics for distributional comparison? How do these metrics handle the complexities of financial time series data, such as fat-tailed distributions or volatility clustering?
> >
> > The L1 distance (or *total variation distance*) is conceptually simple and is proportional to the area of mismatched bins between histograms of both distributions and therefore corresponds to an intuitive measure of distributional similarity. This is also equivalent to the sup-norm between both distributions. However, L1 does not consider *where* both distributions are misaligned and by *how much* on the support of the distributions. This is one advantage of using the Wasserstein-1 distance (earth-mover distance), which produces greater values, the further away the generated distribution's mass lies from the real data distribution. We have added a justifaction of the metrics to the paper to clarify this.
> >
> > Both L1 and Wasserstein-1 can also be applied to heavy-tailed distributions. The issue of volatility-clustering is better addressed by the evaluation of *conditional* distributions, e.g. $p(r_t^2 | r_{t-1}^2)$, which can be performaned with the benchmark and evaluated using either L1 or Wasserstein-1.
> >
> > > **Q4** The benchmark evaluates market impact metrics, but how do you account for outliers or extreme events in real-world LOB data? Are these adequately captured in your current framework?
> >
> > Market impact measures describe LOB behaviour in expectation along the time dimension, as is standard in the literature. Distributional comparison of impact is infeasible as the variance in this noisy financial domain is simply too large, when evaluating impact at each time step. As far as outliers can be in addressed in general, this is done by evaluating entire distributions of scoring functions, including their tails.
> >
> > In principle, model performance with respect to extreme events can be measured in the benchmark framework, for example by calculating the frequency of extreme events as a scoring function. In practice, most current SotA models are however not trained on matching the tail mass correctly, and would therefore likely fail as they already do not have perfect scores with respect to the mass of the distributions that are not in the tail. In the future, adding such a scoring function seems like a good extention of the benchmark though.
> >
> > > **Q5** The conditional evaluation approach looks at conditional distributions of specific LOB statistics. How would this framework perform when dealing with very high-dimensional data, and does it scale well for more complex scenarios?
> >
> > LOB statistics are interpretable examples of our scoring functions $\phi$. The benefit of the scoring functions lies in reducing the dimensionality of the LOB data, so that the distributions can be compared without falling into the sparsity problem due to the curse of dimensionality. The high-dimensional raw LOB distributions would be too sparse to compare between real and generated data. This is exactly why our benchmark framework is designed to handle such complex scenarios.
> >
> > :question: Could you please also give some more details on what exactly you mean by *other complex scenarios*, in case we haven't yet answered this satisfactorily?
> >
> > > **Q6** The paper uses the LOBSTER dataset for evaluation. Given its limitations (e.g., limited asset coverage, certain missing features), do you think the results would generalize to other datasets or more diverse market conditions?
> >
> > LOBSTER provides years of message-by-order (MBO) data for all Nasdaq listed securities, including all buy/sell order, order cancellations, and order executions, and thus describes LOB dynamics on the most granular level during all historical market conditions. We would therefore argue that results derived using this dataset should be generalizable to other data vendors.

---

### Author Response · Authors · 2024-11-22
**General Rebuttal Remarks**

First of all, we must apologise for the clerical error of including a non-anonymised Github link, which was left in the manuscript by mistake at the time of submission. We have fixed this oversight in the updated manuscript and would now kindly ask the reviewers to judge the paper on its merits and help us improve the paper as much as possible and leave the decision regarding the Github link to the AC.

We thank reviewers for some detailed constructive criticism. We hope that we were able to address most concerns satisfactorily and made corresponding changes to the manuscript. We hope that reviewers are now in a position to increase their scores of the paper as we have cleared up some confusion and have aimed to address points of criticism, or to otherwise provide directions for how to further improve the paper.


**Positives**

We thank the reviewers for positive feedback, which we would summarize as follows.

LOB-Bench is judged to offer a nuanced and robust framework for comparison of generative models including *unconditional*, *conditional* and divergence-based evaluation, going beyond just reporting cross-entropy and reproducing stylized facts. The paper also offers the interesting idea of evaluating autoregressive models by binning generated data. Further positives are the inclusion of market impact metrics, well-moticated metrics, and an intuitive and comprehensive visualization of quantitative results. Furthermore, the open-source code enhances reproducibility and practical utility.
It was also lauded that the paper provides the interesting new substantive result, that autoregressive GenAI outperforms for LOB data.

---

### Author Response · Authors · 2024-11-27
**Request for Engagement**

This is a friendly reminder for reviewers **GuKS** and **trTG** to please also *engage* with the rebuttals as the rebuttal process is drawing to a close.

---

### Meta-Review · Area_Chair_L25j · 2024-12-20

**Metareview:**

The paper develops a benchmark to evaluate generative models for limit order books. They also evaluate several models classes including a GAN, parametric model, and an autoregressive model. Two reviewers raise several concerns including the value of the benchmark for future development, the comparison to other benchmarks, and the readability. The question around the utility of the benchmark centered on the question of what the benchmark was powered to detect. Quoting one of the reviewers "Specific experiments that clearly demonstrate superiority in dealing with distribution shift or autoregressive gaps in the generated data would make the method more convincing." These questions were partially addressed in the reply.

**Additional Comments On Reviewer Discussion:**

There was an anonymity violation found by the reviewers.

---

### Decision · Program_Chairs · 2025-01-22

Reject